# “What Do We Know about Hope in Nursing Care?”: A Synthesis of Concept Analysis Studies

**DOI:** 10.3390/healthcare11202739

**Published:** 2023-10-14

**Authors:** Mónica Antunes, Carlos Laranjeira, Ana Querido, Zaida Charepe

**Affiliations:** 1Institute of Health Sciences, Universidade Católica Portuguesa, 1649-023 Lisbon, Portugal; zaidacharepe@ucp.pt; 2Center for Interdisciplinary Research in Health (CIIS), Palma de Cima, 1649-023 Lisbon, Portugal; 3School of Health Sciences, Polytechnic of Leiria, Campus 2, Morro do Lena, Alto do Vieiro, Apartado 4137, 2411-901 Leiria, Portugal; carlos.laranjeira@ipleiria.pt (C.L.); ana.querido@ipleiria.pt (A.Q.); 4Centre for Innovative Care and Health Technology (ciTechCare), Campus 5, Polytechnic of Leiria, Rua de Santo André-66-68, 2410-541 Leiria, Portugal; 5Comprehensive Health Research Centre (CHRC), University of Évora, 7000-801 Évora, Portugal; 6Center for Health Technology and Services Research (CINTESIS), University of Porto, 4200-450 Porto, Portugal

**Keywords:** holistic health, hope, nursing diagnosis, nursing theory, review literature, nurse-patient relations, therapeutical approaches

## Abstract

Hope is a central concept within the nursing literature, which is crucial towards the development of nursing knowledge. Nursing teams play a crucial role in exploring the meaning of hope and promoting hope among patients and their families. This study aims to synthesize concept analysis studies of hope in the context of nursing care and to propose an evidence-based update of the definition of hope in the International Classification for Nursing Practice (ICNP^®^). Method: This is a literature review, involving the synthesis of studies concerning the concept analysis of hope in nursing practice. Peer-reviewed articles with fully accessible Portuguese or English text were considered. As we aimed to include a historical perspective of the concept, no restriction upon the time of publication was set. Articles were selected in March 2022 and updated in July 2023 using the Medline, CINAHL, and Scopus databases. The search terms used were “hope” AND “concept” AND “analysis” AND “nurs*”. Only articles written in English or Portuguese were included. Two reviewers conducted the research synthesis and report independently to minimize the risk of bias in the included studies. This paper adheres to the PRISMA checklist. To clarify the concept of hope as perceived by patients and develop hope as an evidence-based nursing concept, 13 studies were reviewed. The concept of hope, its attributes, antecedents, and consequences, as well as similar concepts, were studied by nurses and synthesized into a definition. The identified antecedents include pivotal life events, stressful stimuli, and experiencing satisfaction with life. The analysis of its attributes, antecedents, and consequences has contributed to understanding its relevance in nursing care and provided a proposed update of hope in the ICNP^®^. This review provides conceptual clarity on how hope is defined and used in nursing practice and the potential factors that may impact the promotion of hope to provide opportunities for future nursing research.

## 1. Introduction

Hope is an integral part of being human and a dynamic and multidimensional phenomenon with numerous meanings. The concept of hope has been studied extensively in many academic fields, including philosophy, psychology, and medicine, although it only appeared in the nursing literature in the 1980s, which defined the concept for specific populations and inspired the first strategies. Since then, hope research and its inclusion in nursing care have become a priority [1]. Hope has become an important and crucial factor in the care of people with health problems and their families [2], with three main focuses: (a) hope as an inner power that facilitates transcending the present situation and conceiving a reality-based expectation of a better future for oneself or others; (b) hope as a state of being, characterized by the anticipation of a good and improved state, or a release from perceived entrapment or a human experience [1,3]; (c) hope as a personal experience, centered on personal responsibility and convictions about the world and our chances of achieving what we want [4].

Scholars agree on describing hope as an idiosyncratic complex process that is essential to life, future-oriented, deliberate, and highly personalized [2,5]. It also entails the presence of substantial goals that are both desired and achievable, as well as the motivation and capacity to make judgments and choices [6].

Dufault and Martocchio provide the most extensive analysis of hope in the nursing literature [7]. They define hope as a dynamic process, a multidimensional “life force” that anticipates achieving a practically attainable and personally relevant good, with some degree of confidence despite marked uncertainty. Within the hoping process, the authors differentiate between generalized hope (a more global longing for some positive, yet undefined, future) and particularized hope (focused on a specific goal) [7,8]. 

Farran et al. [9] conceptualized hope as a psychological and cognitive process that can possess spiritual or transcendent dimensions. It involves the formation of positive expectations and optimistic beliefs about future outcomes [9]. These positive expectations are influenced by various factors, including self-belief (confidence in one’s abilities and potential), trust in others (confidence in the reliability and support of others), and adherence to a religious doctrine (faith in a higher power or spiritual principles) [9].

Hope emerges as a forward-looking process that involves a dynamic interplay among cognition, action, emotions, and interpersonal connections. This process is oriented toward the pursuit of future accomplishments that hold personal significance [10]. 

The International Classification of Nursing Practice (ICNP^®^) defines hope as “the feeling of having possibilities, a trust in others or in the future, a zest for life, an expression of reasons and will to live, and inner peace and optimism, associated with setting goals and mobilizing energy” [11] (p. 1), which can constitute a nursing diagnosis (despair), intervention (promoting hope), or outcome (hope). The continuous updating of this classification ensures its usefulness, relevance, and ongoing quality, as well as supporting future health models and nursing. Regarding the definition of the diagnosis of hope proposed by the ICNP^®^, recent scientific findings point to a better clarification of the concept. In this way, nurses can consider this focus more easily, considering its attributes when making the initial assessment and monitoring nursing care in this area.

Hope research in nursing care has focused on different populations, life spans, and stages of the health–disease transition process [1,12], varying according to the experienced events and cultural environments [13]. Several studies on hope are grounded in nursing’s totality paradigm. In this paradigm, nurses view individuals not just as a collection of isolated symptoms or medical conditions but as whole beings with physical, emotional, psychological, social, and spiritual dimensions. Nurses consider the interplay of these various aspects of a person’s life when providing care, aiming to address the individual’s needs in a comprehensive and integrated manner and are frequently developed to promote healing and quality of life through interventions with specific individuals or certain health conditions [2]. Some examples include elders [14], individuals with chronic medical conditions [15,16], children and adolescents [13,17], caregivers and spouses [17,18], homeless people [19], and people with mental health disorders [20,21].

Recently, there has been an increased interest in the promotion of hope in advanced clinical nursing practice to aid and support clients in managing the complexity of physical and emotional suffering. The nurse’s ability to foster hope necessitates a deliberate approach and the establishment of circumstances for therapeutic interaction through interpersonal engagement. This occurs in response to situations characterized by suffering, uncertainty, or emotional distress, ultimately leading to mutual personal growth for both the nurse and the client [22]. Some of the hope-promoting strategies included making a meaningful impact on their patients’ lives, establishing deep connections with them, accompanying them on their journey, and gradually building trust over time. Additionally, the nurses emphasized that the way they approached their tasks and interactions was more important than the specific actions they took [23]. In this regard, the nursing team plays a crucial role in exploring the meaning of hope and promoting hope in patients and their families. Promoting hope is a professional responsibility and a moral and ethical obligation, which is foundational to all nursing care [24].

There is extensive research on the topic of hope, including several concept analyses, justifying our need to synthesize the evidence, allowing the discipline to attain a clearer comprehension of the concept and contributing to illuminating what “hides in veiled shades of obscurity” [2]. A concept analysis is a valuable approach used in nursing to bring clarity to concepts that are often used with ambiguity. By subjecting these concepts to thorough examination, nurses can achieve a heightened level of insight and a more refined delineation of their role and significance within the scope of their profession. A better understanding of the attributes of a concept makes it possible to increase the accuracy of nursing diagnoses and to target nursing interventions to human responses in health and disease processes. The primary objective of a concept analysis is to identify and explore the essential properties of a concept within the nursing field. Through this process, the aim is to achieve a clearer and more comprehensive understanding of the concept’s meaning and its inherent characteristics, thereby providing a more precise and explanatory framework for its application in nursing practice [25]. A concept analysis, in essence, involves the methodical deconstruction and examination of a particular phenomenon or concept. Undertaking this analytical process allows us to gain a more profound understanding of the concept’s intricacies and nuances, thereby enhancing our comprehension of the essential aspects and significance of quality in bedside nursing care [26].

Therefore, our aims were: (1) to synthesize the research regarding concept analyses of hope in the context of nursing care; (2) to propose an update for the definition of hope in the ICNP^®^.

## 2. Materials and Methods

This focused literature review ultimately aimed to provide an updated definition of hope based on articles that use concept analyses to characterize the concept of hope. This review was conducted and reported independently by two reviewers following the Preferred Reporting Items for Systematic Reviews and Meta-Analyses (PRISMA) guidelines [27].

### 2.1. Review Question

The question of this review is what are the attributes, antecedents, and consequences that define the concept of hope in nursing care? 

### 2.2. Inclusion and Exclusion Criteria

This literature review uses theoretical–methodological models describing the concept of hope in nursing practice. Peer-reviewed articles with fully accessible Portuguese or English text were considered. As we aimed to include a historical perspective of the concept, no restriction upon the time of publication was set. We included concept analyses covering all ages, settings, contexts, and study designs. No filters were established for the year of publication. The exclusion criteria were as follows: (a) grey literature sources, such as theses, dissertations, and conference proceedings; (b) papers on hope or hoping of nurses or other allied healthcare professions, since one of the aims was to contribute to the nursing taxonomy, which requires a patient-centered approach.

### 2.3. Search and Selection Strategy

In March 2022, the authors (M.A., Z.C.) searched four databases: Medline (Medical Literature Analysis and Retrieval System Online), CINAHL (Cumulative Index to Nursing and Allied Health Literature), and Scopus. We updated the search in July 2023. The key Medical Subject Headings (MeSH) terms used included ‘hope’ combined with the free-text keywords ‘concept’ and ‘analyses,’ as well as the truncations ‘nur*’ and ‘hope*’ with the Boolean operator AND. The terms were searched in the title field. The bibliographies of selected articles were searched manually to improve the retrieval of relevant articles. The process of study selection is summarized using a PRISMA flow chart [27] in Figure 1. 

### 2.4. Data Extraction

To eliminate duplicates and aid each reviewer with independent screening, reference manager software was employed. Two reviewers independently read the titles and abstracts of the papers chosen for inclusion. When there was a disagreement or an abstract met the inclusion criteria, it was retrieved for full-text reading. A spreadsheet was shared among the reviewers that included the article’s authors, year of publication, and context; the antecedents, consequents, and definition or attributes of the hope concept; and the empirical referents (Table 1).

A narrative synthesis was used as a methodological strategy to clarify and construct a conceptual model of hope [28].

**Table 1 healthcare-11-02739-t001:** The synthesis of concept analysis studies on hope.

Title/Author/ Date/Country	Concept Analysis Method	Objectives	Antecedents	Attributes/Surrogate Terms/Related Attributes	Consequences	Empirical Referents	Definition
Hope in palliative care nursing: concept analysis Guedes et al. (2021) Portugal [29]	Walker and Avant’s method	To develop hope in palliative care as an evidenced-based nursing concept; to analyze its attributes, antecedents, and consequences	- Positive attitude - Uncertainty - Spirituality - Symptom control - Fatigue - Hopelessness - Interpersonal relationships - Realistic goals/objectives - Depression - Trust - Relationships with professionals - Existential suffering - Request for early death - Nonverbal indications of hope - Keeping busy - Religiosity - Holistic care - Clinical information - Life narrative - Physical condition - Change focus - Preparation for death - Positive events - Diagnosis and chemotherapy time - Hope in healing - Prognostic uncertainty - Resilience	- Positive outcome expectancy: - Inner strength - Coping mechanism - Goal - Spirituality/religion - Expectation - Hope in healing - Prognostic acceptance - Optimism/positive - Living a normal life - Gratitude - Miracle - Security - Protecting patients from distress and suffering - Process-oriented towards the present and the future: - Positive expectancies - Irrational, dynamic, multidimensional, life value - Phenomenon - Living in the moment, generalized, multifaceted, individualized, subjective, attitude	- Stress reduction - Resilience - Quality of life - Peaceful death - Life extension - Improvement - Legacy - Positive future for family and friends - Survival - Acceptance - Holistic care - New meanings and objectives - Future gains	“Adherence to therapy, a sense of well-being, having a positive attitude, and the ability to engage in self-management, despite physical limitations” (p. 341). Also: The Hope Communication Tool, Beck Hopelessness Scale, Snyder Hope Scale, Herth Hope Index, Hopelessness Assessment in Illness, Miller Hope Scale, Nowotny Hope Scale.	“Hope is central to the adjustment process in palliative care when trying to maintain a sense of normalcy and developing cognitive, social, behavioral, and transcendental strategies to improve confidence” (p. 342)
A Critical Analysis of the Concept of Hope: The Nursing Perspective Nweze, Agom, and Nwankwo (2013) Nigeria [30]	Walker and Avant’s method	“To examine the concept of hope within the healthcare context, with a particular focus on hospital nursing as presented in the literature” (p. 1027).	- Suffering, pain, and despair brought about by chronic and debilitating diseases - Pivotal events of life - Stressful stimuli such as loss, hardship, and uncertainty - Positive personal attributes - Connectedness with God - Acknowledgement of a threat	- Spirituality - Goals - Comfort - Caring - Interpersonal relationships - Control - Expectation - Life review	- Ability to achieve goals - Being able to cope and experiencing life satisfaction despite the limitations brought about by illness	“Adherence to medication, sense of well-being, good positive attitude, and being able to engage in the management of self despite physical limitations” (p. 1029).	“Hope is experienced within the hospital culture, events, and interactions that occur between patients and healthcare givers can provide the meaning or understanding of what happens in current practice” (p. 1030).
Hope and Parents of the critically ill new-born: a concept analysis Amendolia (2010) USA [31]	Walker and Avant’s method	To examine the concept of hope in parents of critically ill new-borns	- Negative and stressful stimuli - Suffering or loss - Predicament or threat - Uncertainty	- Future orientation - Goal setting - Realism - Energy or activity processing - Positive feeling or optimism	- Ability to cope - Certainty - Improved health - Improved quality of life - Peace - New perspective - Strength - Empowerment	The Herth Hope Scale (HHS), the Hope Index Scale, the Childrens’ Hope Scale, the Stoner Hope Scale, the Miller Hope Scale, and the Nowotny Hope Scale (p. 143). The Hope Assessment Guide was developed to indicate the progression and development of hope.	“Hope plays a role in the day-to-day experiences of families in this setting and can improve the overall experience at such a stressful crossroad. Fostering and maintaining hope should be a priority for those entrusted with the care of the critically ill new-born” (p. 144).
Hope in early-stage dementia: a concept analysis Cotter (2009) USA [32]	Walker and Avant’s method	“To explore hope in early-stage dementia and to identify the dynamics and components of the hope experience” (p. 232)	- Awareness or recognition of the diagnosis - Managing a sense of self in the context of relationships and social identities	- Hope in the experience of loss and ongoing adjustment - Future orientation of hope - Hope and social identity and social network - Hope and adaptation to daily living (p. 233).	- Positive changes in the self-concept -Growth opportunities.	“Hope is a process occurring within the individual, whose outcomes can be measured by self-report; clinical scales; and observations of body language, posture, facial expressions, and behavior” (p. 236).	“Hope is central to the adjustment process in early-stage dementia when trying to maintain a sense of normalcy and developing cognitive, social, and behavioral strategies to improve confidence. Maintaining hope, helping others, and living within a supportive social network can positively influence adaptations to daily living and to the preservation of self-concept” (p. 235).
An exploration of hope as a concept for nursing Tutton, Seers, and Langstaff (2009) United Kingdom [26]	Morse’s pragmatic utility approach	To examine the concept of hope within the context of healthcare, with a focus on hospital nursing, as portrayed in the literature	Not provided	- Hope as an expectation for the future - Hope as a cognitive process - Goal attainment	Not provided	Not provided	Hope is classified as an expectation or a cognitive process encompassing realistic and unrealistic hope.
Paradox of Hope in Patients Receiving Palliative Care: A Concept Analysis Tanis, DiNapoli, and Hampshire (2008) [33] UK	Walker and Avant’s method	To conceptually define the paradox of hope, to operationalize the concept, and to apply the concept to nursing practice	- Certainty vs. uncertainty - Comfort vs. discomfort - Goals vs. not caring - Peaceful vs. grieving - Acceptance vs. struggle - Control vs. lack of control - Companionship vs. abandonment - Joy vs. depression Reaching for life vs. acceptance of dying	- Dynamic quality along a continuum - Personal experience of hope - Continuous present	- Coping skills - Support and companionship - Value of the individual - Personal relationship with a higher power - Healthcare providers understand hope	The structured interview assessment of symptoms and concerns, Beck Depression Inventory-II. Hospital Anxiety and Depression Scale, and Herth Hope Index (HHI).	Hope is the lived experience of the patient searching for meaning in life, engaged in a continuous dialogue about the consequences of dying. As a result of this dialogue, attitudes change, choices are considered, and life is reshaped
Hope: more than a refuge in a storm. A concept analysis using the Wilson method and the Norris method. Sachse (2007) USA [34]	Wilson’s conceptual model	To provide an operational definition of hope	- Genetic temperament - Scripting from significant others - Experiences (personal and observed) - Memories, beliefs, and values - Wished for object - Dilemma - Crisis	- Universal yet unique to each individual - Dynamic in its presence - Enabling	- Resilience - Transcendence - Positive psychologically, spirituality, physiologically	Not provided	Hope is defined as “a multidimensional construct arising from our memories, beliefs, and values” (p. 1552); hope is part of all activities and thoughts that strengthen the spirit, facilitating behavior to elicit an outcome or promote a level of comfort while impacting life quality.
Hope in terminal illness: an evolutionary concept analysis. Johnson (2007) New Zealand [35]	Rodgers’ evolutionary model	To clarify the concept of hope as perceived by patients with a terminal illness, to develop hope as an evidence-based nursing concept, and to contribute new knowledge and insights about hope to the relatively new field of palliative care, endeavouring to maximize the quality of life of terminally ill patients in the future	- Being physically unwell, pain, discomfort - A medical diagnosis of a terminal illness - Uncertainty - Fear of dearth - Feelings of devaluation of personhood	- Personal qualities (such as inner strength, determination, and an optimist state of mind) - Spirituality - Goals - Comfort - Help/caring - Interpersonal relationships - Control - Legacy - Life review - Positive expectations	- Feeling calm, peaceful, and accepting the situation - Problem-solving approach - Coping mechanism - Optimize the quality of life	Not provided	“A working definition of hope in terminal illness is that it is a calm, emotional hope that is global in nature, focusing on the hopes of loved ones rather than on a fulfilling, prosperous future. Hope is living in the present and spending quality time with significant others. An enrichment of being is more important than having or doing. Hope is directed towards comfort, peace, leaving a legacy, and considering spiritual dimensions. Hope is about valuing the gift of each day with the positive expectation of a few more good days to follow. Patients review the life they have led and attach meaning to their past achievements, which equates to hope and maximizes quality of life.” (p. 458)
One step towards the understanding of hope: a concept analysis. Benzein and Saveman (1998) Sweden [36]	Walker and Avant’s method	To elucidate the concept of hope	Stressful stimuli - Loss - Life-threatening situations - Temptation to despair	- Future orientation - Positive expectation - Intentionality - Activity - Realism - Goal setting - Interconnectedness	- Ability to cope - Renewal - New strategies - Peace - Improved quality of life - Physical health	Not provided.	“Hope is a human phenomenon with no sharp boundaries, related and interwoven with other phenomena, e.g., expectations and desire. Hope can be seen as one link in a field of emotions such as hope–joy–enthusiasm or expectation–yearning–confidence–hope. Hope can be seen as a personal experience; the only true understanding comes from sharing people’s narrated experiences of hope” (pp. 327–328).
An analysis of the concept of hope in the adolescent with cancer. Hendricks-Ferguson (1997) USA [37]	Walker and Avant’s method	“To provide a sample concept framework for paediatric oncology nurses that will assist them in applying the conceptual attributes in their clinical practice with adolescents diagnosed with cancer” (p. 73).	- Previous experience in trusting and loving relationships with significant others - Previous experience with successful learning experiences - A history of successful goal obtainment - A stressful stimulus, such as being diagnosed with cancer (p. 77)	- Positive thinking or optimism - Reality-based and future-oriented goals - Positive future for self or others - Positive support systems	- The individual expresses new feelings of safety or comfort - The individual formulates new and realistic goals - The individual demonstrates a belief that what is hoped for is possible - The individual expresses confidence in the future - The individual expresses a concern for and a focus on others in addition to self The individual conveys trust in supportive actions from others (p. 77).	The Hopefulness Scale for Adolescents (HAS) is a 24-item, visual analogue scale designed to quantify the degree of positive future orientation an adolescent feels at the time of measurement. “A short interview or using an existing spiritual well-being assessment scale in combination with the HSA is recommended because of the HSA’s lack of any items related to spiritual well-being” (p. 78).	The theoretical construct of adolescent hopefulness is defined as “the degree to which an adolescent possesses a comforting or life-sustaining, reality-based belief that a positive future exists for self or others” (p. 77). The concept of hope for the adolescent is currently in the early stage of development and understanding.
Simultaneous concept analysis of spiritual perspective, hope, acceptance, and self-transcendence. Haase et al. (1992) USA [25]	A simultaneous concept analysis based on the Wilson method and described by Walker and Avant	“To clarify the four concepts simultaneously, offering mutually exclusive theoretical definitions for each concept while highlighting their interrelationships and distinguishing characteristics” (p. 141).	- A pivotal life event or a stressful stimulus such as loss, major decisions, hardship, suffering, and uncertainty - Positive personal attributes - Connectedness with others or God, a feeling of uncertainty, uneasiness, or other related feelings of discomfort (p. 143)	- Future-oriented - An energized, action orientation termed variously as activity - Either a generalized or particularized goal or a desired outcome	- A sense of personal competency in meeting goals - A ‘winning position’ - Peace - Ability to transcended	Not provided.	Hope is “defined as an energized mental state involving feelings of uneasiness or uncertainty and characterized by a cognitive, action-oriented expectation that a positive future goal or outcome is possible” (p. 143). Clearly, spiritual perspectives, hope, acceptance, and self-transcendence are just three of the many dynamic psychosocial processes requiring further theoretical and empirical attention.
The concept of hope revised for nursing Stephenson (1991) USA [38]	Walker and Avant’s method	To review definitions and conceptual usages of the word “hope” from the literature and answer the conceptual question of “what is hope?” (p. 1456)	- Crises (could include a loss, a life-threatening situation, a hardship, or a change - A difficult decision or challenge - Anything that would be significant to the person	- The object of hope is meaningful to the person - Hope is a process involving thoughts, feelings, behaviors, and relationships - There is an element of anticipation - There is a positive future orientation, which is grounded in the present and linked with the past” (p. 1459)	- New perspective (hope seems to energize, empower and strengthen the person) - People with their hopes fulfilled, describe themselves as invigorated, full of purpose, renewed, calm and encouraged” (p. 1459).	Not provided.	“From these definitions and attributes, a tentative definition of hope can be proposed; hope is an anticipation, accompanied by desire and expectation, of a positive possible future” (p. 1457).
Hope in elderly with chronic heart failure. Concept analysis Caboral et al. (2012) USA [39]	Walker and Avant’s method	“To explore the construct of hope in elderly adults with chronic heart failure” (p. 406)	- Suffering and despair brought about by heart failure illness (decreased functional capacity)	- Future orientation: The future in older adults is the anticipation of a future in each day lived, and they look at the past instead of the nurture. Past successes nurture their hope. They hope that their condition does not worsen - Sense of limitation: “The HF illness trajectory can be unpredictable and range from being able to function without difficulties to a severely limited functional capacity. Hopeful individuals with HF maintain their involvement in life despite the limitations imposed by the illness trajectory” (p. 408)	- The ability to achieve goals “Goals in HF include being able to cope and experiencing life satisfaction despite limitations brought by the illness” (p. 410)	“Adherence to therapy, a sense of well-being, having a positive attitude, and the ability to engage in self-management despite physical limitations could be empirical referents of hope” (p. 410)	“Hope is an intangible concept that is difficult to observe and is imbedded within someone’s personal experience. It is a belief that something positive without any guaranteed expectation that it will occur” (p. 410)

### 2.5. Data Analysis

The studies were synthesized separately by two researchers while considering the main categories needed for the synthesis—antecedents, consequents, and attributes. To ensure the study’s validity and reliability, the procedure was overseen by researchers acquainted with this method of analysis.

## 3. Results

A total of 322 articles were initially identified. After removing duplicates, 218 titles or abstracts were excluded were deemed irrelevant, resulting in 17 articles whose full text was read. Four of these articles were excluded as they did not comply with the inclusion criteria; specifically, they did not use a conceptual analysis or evaluate other concepts (e.g., hopelessness, resilience, spirituality). In total, 13 articles were included after data extraction. 

Those articles were published from 1991 to 2021 in English. The theoretical–methodological model used most often was Walker and Avant’s [40] concept analysis model [29,30,31,32,33,36,37,38,39]. Among other models used were the evolutionary approach of Rodgers [35,41], Wilson and Norris’ method [25,26,27,28,29,30,31,32,33,34,35,36,37,38,39,40,41], and Morse’s concept analysis [26].

Different populations and clinical settings were included in this analysis: palliative care from Portugal [29] and the United Kingdom [33]; patients with a terminal illness from New Zealand [41] and Nigeria [30]; adolescents with cancer from the USA [37]; parents of critically ill newborns [31]; psychiatric population [35]; patients with early-stage dementia [32]; families and pediatric caregivers during a child’s serious illness [26]; orthopedic and trauma settings from the United Kingdom [25]; elderly adults with chronic heart failure from the USA [39] (Table 1).

Some authors [36,38,39] have elucidated the concept of hope with a review of definitions and conceptual uses. Additionally, another author provided mutually exclusive theoretical definitions of four concepts simultaneously—spirituality, hope, acceptance, and self-transcendence—highlighting their interrelationships and distinguishing characteristics [34]. In the process of result formulation, we conducted a review of the antecedents, attributes, and consequences of hope (Figure 2). For the antecedents, we identified variables that precede or influence hope, such as life events, personal characteristics, or environmental factors. Regarding the attributes, we established clear definitions and quantified the constituent elements or features that comprise the construct of hope, including agency, pathways of thinking, and emotional experiences. The consequences of hope determined the outcomes and effects associated with hope, encompassing both positive and negative manifestations [22].

Figure 2 illustrates the link between antecedents, attributes, and consequences in a synthesis model, resulting in two main dimensions: “hope is a dynamic emotional state” and “hope is central to the process of adaptation”. These dimensions highlight that hope is not a static emotion (dynamic emotional state); it changes over time and plays a central role in how individuals adapt to and navigate the challenges and uncertainties of life (adaptative process). For this reason, it serves as a motivating and adaptive force that can help people persevere and thrive in various situations.

### 3.1. The Antecedents of Hope

The data indicate that hope is a comprehensive concept that requires support to achieve positive outcomes [39,42]. Antecedents of hope may consist of a pivotal life event or a stressful stimulus, such as loss, major decisions, hardship, suffering, pain, despair, and uncertainty due to illness, chronic diseases, or terminal illness. Antecedents of hope may also include a predicament or threat [29,30,31,33,34,36,38,39,41], such as an uncertain prognosis, experience of a trusting and loving relationship [26,29,41], a successful learning experience, reaching a goal [32,35,37], and connectedness with others and God or spirituality [29,30,32,34]. Antecedents such as “trust, interpersonal relationships, effective communication, and relationships with health professionals” underline the importance of holistic care [30,41] (p. 1029). It is crucial to consider and prioritize these aspects, as they play a vital role in the overall well-being and satisfaction of patients or individuals receiving healthcare services.

### 3.2. The Attributes of Hope

Hope was defined and described in several ways: an expectation of positive outcome [25,29,31,33,36,37,38,41]; an orientation towards the present and future [26,36]; a multidimensional and dynamic process of goal identification [25,26,29,31,33,34,36]; a personal quality [35,36,37,38,41]; a cognitive decision-making process [25]; spirituality [26,41]; a feeling of comfort and safety [30,41]; an interpersonal relationship and feeling of belonging and being needed [36,41]; a legacy [41]; an energy–action towards the future [26,31,34,36]; an ongoing adjustment to personal and observed experiences during illness and loss to daily living [32,33,35]; a sense of limitation [39]; a coping strategy [26,29,37,38]; mutuality experiences [41]; a positive social support [37]. 

These attributes frame a more pragmatic view (see the Appendix A) and can help understand how nurses might identify hope in the continuum of care.

### 3.3. The Consequences of Hope

Possible consequences of the concept indicated in the literature included the following: returning to a more stable life, with better physical health and quality of life [26,30,31,32,33]; finding hope and meaning in life [29,34,37,38]; overcoming a difficult situation and obtaining greater healing and satisfaction [29,38]; facing uncertain future constructively [29,36,38,41]; accepting a health condition [29,30,34,36,38,41]; coping with pain and suffering [29,30,34,36,38]; promoting a sense of renewal, peace, and legacy, and developing new strategies and a positive outlook for family and friends [29,36,38]; survival [29,35,36,38]; positive changes in self-concept and growth opportunities [29,30,32,34]; developing new feelings of safety or comfort, formulating new and realistic goals, and promoting confidence in the future and trust in the support from others [30,34,37,39,41]; faith and spirituality [35]; a winning position [34]. Faith and the perception of the transcendent emerge as consequences, as they become resources for individuals, particularly in times of uncertainty and vulnerability. 

### 3.4. Updating the Definition of Hope

The authors compared classifications (NANDA international diagnoses, ICNP^®^ axis, and similar terms) and proposed a new definition for the hope diagnosis and a new term for the ICNP^®^ (Table 2).

The data suggest that is possible to update the definition of hope in nursing practice in the ICNP^®^ [11], providing a new insight into the relationship between hope and nursing practice and focusing on how people endure and sustain a positive outlook on life, despite their difficulties. Compared to the ICNP^®^ definition, which is more concise and emphasizes certain emotional aspects of hope, our proposed definition of hope is more detailed, multidimensional, and contextually flexible. Our proposed definition also explicitly includes a cognitive element and highlights the connection between hope and the present, as well as the future.

## 4. Discussion

This analysis supports the theory that hope is a multidimensional concept, described as an emotion, state, and experiential process [10]. Based on the concept’s antecedents, attributes, and consequences, some authors formulated a conceptual definition. From a chronological perspective, several definitions have arisen: (a) hope as the anticipation of a positive possible future, accompanied by desire and expectation [38]; (b) hope as an energetic cognitive state, an action-oriented expectation of a positive future aim or outcome [25]; (c) hope as dynamic, multidimensional energy that promotes positive expectations, motivating actions to attain personally significant and realistic future goals [43]; (d) hope as the reassuring or life-sustaining, realistic belief in a positive future for self or others [31]; (e) “it is a belief in something positive without any guaranteed expectation that it will occur” [25] (p. 410); (f) hope in terminal illness represents “a calm, emotional, global focus on the hopes of loved ones” [42] (p. 458); (g) hope is “a multidimensional construct arising from our memories, beliefs, and values”, which permeates all our activities and thoughts, strengthening the spirit and facilitating behavior that promotes the desired outcome or level of comfort, while impacting life quality [44]; (h) hope is an expectation of a sustained good and improved state, or a release from perceived entanglement [45]; (i) hope is crucial to the adjustment processes in health–illness transitions [34,39]. 

Hope is a process with individual value derived from lived experiences [2]. Some authors report behaviors highlighting aspects such as self-therapy [34], medication adherence, a sense of well-being, a positive attitude, and self-management despite physical limitations [34]. The findings from studies indicate that hope plays a crucial role in sustaining determination and resilience, even in the absence of tangible positive evidence. Additionally, hope can broaden an individual’s perspective, opening new possibilities and potential pathways for coping and achieving positive outcomes [46]. Consequently, these findings would provide valuable empirical support, offering caregivers the opportunity to refine their abilities in fostering and strengthening hope in individuals during their routine interactions and experiences.

The empirical referents identified in six studies are helpful for decision-making in nursing care. Hope has been measured both qualitatively and quantitatively. From a qualitative perspective, the interview is the most used method to understand the experience of hope in different populations. From a quantitative perspective, the research has used relevant hope assessment instruments, including the Hope Communication Tool, Beck Hopelessness Scale, Snyder Hope Scale, Hopelessness Assessment in Illness, Children’s Hope Scale, Stoner Hope Scale, Hope Assessment Guide, Beck Depression Inventory-II, Hospital Anxiety and Depression Scale, and Hopefulness Scale for Adolescents. 

The instruments used most often in the analyzed studies were the Herth Hope Index, Miller Hope Scale, and Nowotny Hope Scale. It should be noted that some of these instruments do not meet the gold standards of psychometric evaluation [45]. For that reason, is required to improve the quality of available hope scales and adapt them to different cultural scenarios. 

The obtained results build upon evidence from conceptual analyses to clarify the concept of hope. This approach should be applied further in nursing research, contributing to a clearer understanding of how to improve hope in vulnerable people in clinical settings. In this sense, advocating and fostering hope might be a useful resource that allows people to maintain or regain their well-being [47,48]. 

This review contributes to advancing the comprehension of the concept of hope. This study’s antecedents, critical attributes, and consequences of hope offer valuable insights that can benefit nurses working in diverse healthcare environments. By utilizing this knowledge, nurses can more effectively recognize and grasp the experience of hope in their patients, thereby enhancing their ability to provide optimal care and support.

### 4.1. Strengths and Study Limitations

The narrow focus of the topic, the exhaustive search for evidence, the criterion-based selection of relevant evidence, and the objective summary offered for the conceptual hope map are all strengths. 

The review was not conducted using a systematic approach, and the gray literature, unpublished academic articles, and editorials were not included. Additionally, a small number of databases were used to identify suitable research.

The results obtained need to be framed with caution, even though they provide important information for clarifying the concept of hope. Building a synthesis matrix to represent an ever-expanding and complex set of literature requires some degree of simplification and subjectivity. Therefore, this overview may evolve for new insights and should serve as the first step toward a more complete and comprehensive understanding of hope.

The presence of hope, even in stressful and life-changing events, often leads to a sense of control, optimism, and a new purpose in life. However, having false hope about possible outcomes and denial of reality as a coping mechanism because one cannot accept reality is an unrealistic way to improve the situation. It is, therefore, vital to learn to distinguish between potentially feasible and impossible goals of self-change to avoid overconfidence and false hopes that ultimately lead to failure and suffering. In some of the studies reviewed, unrealistic hope is essential to consider when evaluating hope, requiring further research through a more widespread understanding. Additionally, it could be worthwhile to offer an interdisciplinary overview of different perspectives of hope across the different scientific disciplines.

### 4.2. Implications for Nursing Practice and Research

Applying this review’s results to real-life decision-making in care management is of central importance. This may result in specific nursing procedures to assess the existence or absence of hope in distinct groups. Thus, nurses might help clients discover and build successful hope-inspiring strategies [6], as well as achieve their desired abilities and goals. Nurses can take advantage of reflection on their clinical practice to identify their viewpoints on hope and how they incorporate hope-inspiring approaches into their professional work. These reflections could play a vital role in fostering fresh or enriched perspectives on hope through descriptive analyses, self-awareness, emotional examination, critical thinking, syntheses, and evaluations. 

The significance of hope in nursing practice is well-established, extending beyond a professional responsibility to a moral and ethical obligation for all nurses. Nursing curriculums should include teachings on hope, its assessment, and how to integrate hope-inspiring strategies into patient care plans. Clinical experiences should focus on empathetic connections with diverse clients, using the term “hope” comfortably. The proposed hope process model can serve as a framework for assessing hope and selecting appropriate nursing strategies based on the client’s phase of hoping [49]. In advanced nursing education, a comprehensive approach to promoting hope should be adopted. This could involve incorporating hope intervention programs into clinical practicums and program evaluation courses to assess their effectiveness. Additionally, conducting doctoral research on the phenomenon of hope could yield valuable insights for future studies, contribute to the development of knowledge, and enhance evidence-based nursing practices.

This hope synthesis review is also a significant subject for nursing research. Despite the small number of studies analyzing the concept of hope, the knowledge already developed around its attributes, antecedents, and consequences is relevant. For this reason, it is important to conduct clinical trials to evaluate the effectiveness of interventions and activities that promote hope [6]. The synthesis of the various components of the concept of hope raises the opportunity to construct assessment instruments that measure its attributes. The continuity of research about the concept of hope in nursing care and the increase in its conceptual understanding will influence methodological decisions, as well as the boundary variables, in experimental clinical studies. These studies will validate the proposed conceptual model.

The research should also focus on hopefulness mediators, such as general well-being, social support, spiritual well-being, coping responses, gender differences, and cultural issues. Given that most studies occurred in so-called “Western” countries [8], future studies should examine unexplored contexts and consider how different aspects of hope have a role in different settings, cultures, and groups (e.g., obstetric care settings, prison contexts, community settings, and ghettos). 

## 5. Conclusions

This study’s primary purpose was to synthesize prior research components on concept analyses of hope in nursing and provide an updated definition, broader and more contemporary antecedents and attributes, and more specific consequences of hope in nursing than earlier concept analyses. This synthesis has provided detailed information regarding the causes, characteristics, and effects of hope, as well as vital insights into the needs of various groups and a deeper understanding of the idea of hope. 

Overall, the review emphasizes hope as a multifaceted construct with emotional, cognitive, motivational, social, and identity-related components, bringing consolation when facing life’s challenges. 

A better operationalization of the concept of hope can be achieved from the synthesis of concept analyses. Reducing ambiguity regarding its understanding will allow nurses to progress in measuring and selecting activities within the promoting hope intervention. Studying the activities and interactions between patients and healthcare professionals and how hope is experienced in healthcare settings could help clinicians understand what is currently happening in healthcare practice and show how this relates to the identified aspects of hope. This, in turn, could help the nurses improve healthcare workers’ abilities to preserve and foster hope in others. By employing evidence-based research, nurses could effectively anticipate potential outcomes and use appropriate interventions to nurture and bolster patients’ hope during nursing care.

Hope is a central concept in nursing, directly affecting nursing practice in patient care and nursing education. Although considered alongside other adjacent concepts, hope has its theoretical structure and measurement instruments and is definable based on knowledge in the scientific nursing literature. This review provides a better grasp of how hope is defined, actualized, and employed in nursing practice and the possible circumstances that may provide opportunities and promote hope.

## Figures and Tables

**Figure 1 healthcare-11-02739-f001:**
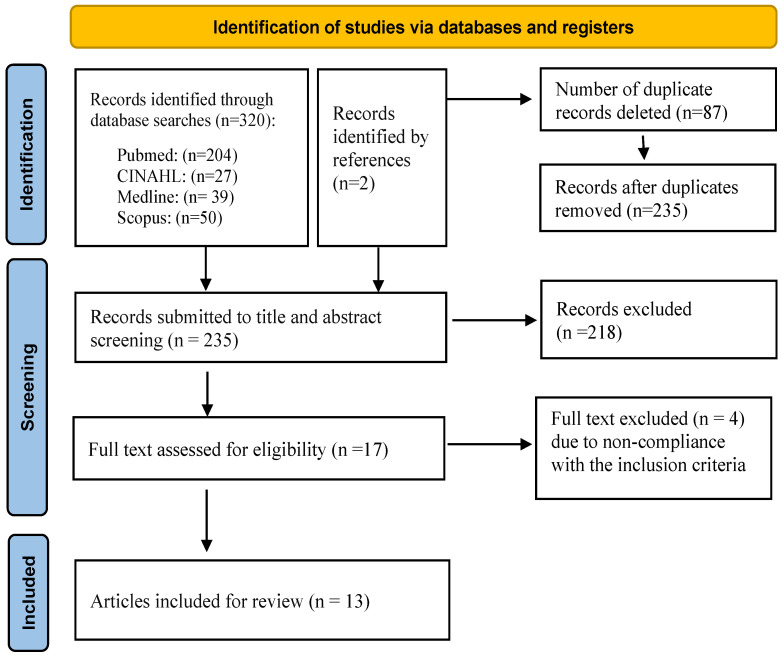
PRISMA flow chart of the study selection [27] (adapted to https://www.equator-network.org/reporting-guidelines/prisma/, accessed on 1 June 2023).

**Figure 2 healthcare-11-02739-f002:**
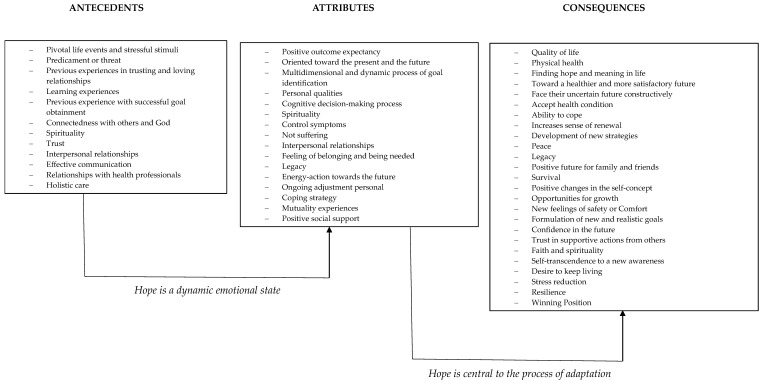
The synthesis model of hope.

**Table 2 healthcare-11-02739-t002:** Proposed update of the definition of hope in the ICNP^®^.

Definition of Hope—Code 10009095 (ICNP, 2019)	Proposal of Hope Definition for ICNP^®^
Emotion: Feelings of having possibilities, trust in others and the future, zest for life, expression of reasons and will to live, inner peace, optimism, associated with setting goals and mobilization of energy.	A dynamic emotional state, multidimensional energy that evokes a positive outcome expectancy and is process-oriented toward the present and the future. Depending on the context, hope can focus on living a fulfilling life around important people, legacy, spiritual dimensions, and maximizing quality of life. Hope is central to adapting to uneasiness or uncertainty. It is characterized by a cognitive, action-oriented expectation that a positive future goal or outcome is possible.

## Data Availability

Materials are available from the corresponding author upon request.

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
