# Peer review of "“What Do We Know about Hope in Nursing Care?”: A Synthesis of Concept Analysis Studies"

_healthcare, 2023, doi:10.3390/healthcare11202739_

Round 1

Reviewer 1 Report

Dear authors

Thank you for submitting the paper  "What do we know about hope in nursing care?": a systematic synthesize review of concept analysis studies" for appreciation at the Healthcare journal.

The paper contributes significantly to the advancement of nursing practice and the concept of Hope. It provides relevant and timely information to support the optimization of client care. 

Please clarify in the Introduction why there is a need to update the concept of Hope in the ICNP, and if the new concept will be adopted by the ICNP.

All the best,

Author Response

We wish to express our sincere appreciation for the invaluable suggestions offered by you in your role as a reviewer for our article titled “What do we know about hope in nursing care?: A systematic synthesis review of concept analysis studies.” We are committed to providing a point-by-point response to your feedback. On behalf of the authors, we extend our heartfelt thanks for granting us the opportunity to review and provide clarification on these crucial matters.

REVIEWER 1

Reviewer 1: "Please clarify in the Introduction why there is a need to update the concept of Hope in the ICNP, and if the new concept will be adopted by the ICNP."

Response to Reviewer 1: Thank you for reviewing our article. This question has been explained with track changes in the main article. The Classification of Nursing Practice aims to standardize concepts and catalogue nursing diagnoses, outcomes, and interventions, creating a common terminology for all nurses. The continuous updating of this classification ensures its usefulness, relevance, and ongoing quality, as well as supporting future health models and nursing. About the definition of the diagnosis of hope proposed by the ICNP®, recent scientific findings point to a better clarification of the concept. In this way, nurses can consider this focus more easily, considering its attributes when making the initial assessment and monitoring nursing care in this area. Given that contact with the ICN had not yet been established in this review, we removed this intention from the article.

Reviewer 2 Report

 This is a systematic synthesis review of concept analysis studies study and explores the hope in nursing care. The following suggestions and questions are intended to help the authors revise this paper for future publication.

Introduction:

1.      There are several paragraphs with only one sentence (Line 57-58, Line 66-73) or two sentences (Line 47-49) that is not adequate in academic writing.

2.      Suggestion: Revisions should focus on improving the logical content. For instance, the flow of origination, definition, and exploration of more specific concepts/issues related to hope would be better. The existing content flow in the introduction is mixed up with origination, definition and other issues of hope . This could confuse readers while reading.

3.      There is an excessive use of direct citations “… ”  which is fine for definitions, but not for other issues. Suggestion: authors can use your words to present the ideas and information from the literature instead of direct citations “… ” .

4.      Line 42: Please clarify the term of The concept here.

5.      The content in Line 41 and Line 614 is duplicated?

6.      Line 80: Please explain what the nursing’s totality paradigm is.

7.      Line 99-101: please clarify the meaning or grammatical issues. Using so and thus in one sentence.

8.      Line 104-105: Meaning unclear- ….nurses can gain a better understanding and more precise definition of their meaning in the context of their profession.

Materials and Methods

9.      Suggestion: need to improve the methodology of concept synthesis in this paper. The concept synthesis has a methodology foundation that helps to clarify concepts of definitions and identify the types of evidence within a given field or topic. Add the concept synthesis methodological literature needed.

10.  Line 134-135: mention the exclusion criteria b) is papers on hope or hoping of nurses. However, the title of the paper included is “A Critical Analysis of the Concept of Hope: The Nursing Perspective”. Please clarify or edit the contents.

11.  Figure 1 PRISMA flow chart: please reconfirm the number. In the identification step, there are 320 plus 2 articles and then records after duplicates are removed (n=234). Why did the screening step show N=235?

12.  The format of Figure 2 needs to adjust (Line 187-188, 197-220, 203-205).

13.  Please explain how to form the results of “hope is a dynamic emotional state” and “hope is central to the process of adaptation” in Figure 2.

Results

1.      Suggest: Authors need to explain in detail how to form the results of antecedents, attributes and consequences of hope in this paper. There is no information provided.

2.      Suggest: Table S1 should be in the context because it provides crucial information of this manuscript.

3.      Generation of new knowledge related to hope from this manuscript is critical. add and explain in detail what the updated definition of hope is compared with the ICNP. The information provided here is not sufficient.

4.      Line 210: mention these qualities suggest…. Please clarify the meaning of these qualities here.

5.      Line 217-218: if the direct citation “… ” is needed here, authors should explain their meaning.

6.      Line242-243: Please explain the results of faith and spirituality, and winning position. The previous consequences of hope have a verb such as developing a new feeling of safety and formatting new and realistic goals.  Only using the terms of faith and spirituality, and winning position as the consequences of hope do not make sense.

 Discussion:

Suggestion: Given a heading of antecedents, attributes and consequences of hope and discuss in each would be great.

Author Response

We wish to express our sincere appreciation for the invaluable suggestions offered by you in your role as a reviewer for our article titled “What do we know about hope in nursing care?: A systematic synthesis review of concept analysis studies.” We are committed to providing a point-by-point response to your feedback. On behalf of the authors, we extend our heartfelt thanks for granting us the opportunity to review and provide clarification on these crucial matters.

REVIEWER 2

Reviewer 2: Introduction Section_"There are several paragraphs with only one sentence (Line 57-58, Line 66-73) or two sentences (Line 47-49) that is not adequate in academic writing.”

Response to Reviewer 2: Thank you. We have increased the number of sentences in each paragraph.

Reviewer 2: Introduction Section_Suggestion:"Revisions should focus on improving the logical content. For instance, the flow of origination, definition, and exploration of more specific concepts/issues related to hope would be better. The existing content flow in the introduction is mixed up with origination, definition and other issues of hope. This could confuse readers while reading”.

Response to Reviewer 2: Thank you for your suggestion. The content of the "Introduction" section has been revised to include the following structure: flow of origin, definition, and exploration of more specific concepts/issues related to hope. Only the most important changes have been highlighted due to the amount of changes made in this section.

Reviewer 2: Introduction Section_“There is an excessive use of direct citations “…”  which is fine for definitions, but not for other issues. Suggestion: authors can use your words to present the ideas and information from the literature instead of direct citations “…”

Response to Reviewer 2: We sincerely appreciate your relevant suggestion. We have now replaced direct citations with our own language throughout the text, with the exception being definitions where citations are retained.

Reviewer 2: Introduction Section_“Line 42: Please clarify the term of the concept here.”

Response to Reviewer 2: We have incorporated the referenced concept into the text in continuity with the restructuring carried out in this section. Thanks again for your suggestion.

Reviewer 2: Introduction Section_"The content in Line 41 and Line 614 is duplicated?”

Response to Reviewer 2: Dear Reviewer, we cannot find line 614 in the article you are reviewing. This seems to be an oversight, and we ask that you correct it so that we can answer this question.

Reviewer 2: Introduction Section_"Line 80: Please explain what the nursing’s totality paradigm is.”

Response to Reviewer 2: We have incorporated the referenced explanation into the text.

Reviewer 2: Introduction Section_"Line 99-101: please clarify the meaning or grammatical issues. Using so and thus in one sentence.”

Response to Reviewer 2: We have addressed and resolved the meaning and grammatical concerns.

Reviewer 2: Introduction Section_"Line 104-105: Meaning unclear … nurses can gain a better understanding and more precise definition of their meaning in the context of their profession.”

Response to Reviewer 2: We have provided elucidation for the sentence in question as mentioned Line 114-118. 

Reviewer 2: Material and Methods Section_"Suggestion: need to improve the methodology of concept synthesis in this paper. The concept synthesis has a methodology foundation that helps to clarify concepts of definitions and identify the types of evidence within a given field or topic. Add the concept synthesis methodological literature needed.”

Response to Reviewer 2:  We sincerely appreciate your relevant suggestion. This article aims to systematically synthesize research regarding concept analyses of hope in nursing care and to propose an evidence-based update of the definition of hope. For this reason, no concept analysis methodology was detailed. In the results section, the articles included in this review followed different concept analysis methodologies, which is why we consider that it constitutes a characterizing item and not the methodology followed.

Reviewer 2: Material and Methods Section_"Line 134-135: mention the exclusion criteria b) is papers on hope or hoping of nurses. However, the title of the paper included is “A Critical Analysis of the Concept of Hope: The Nursing Perspective”. Please clarify or edit the contents.”

Response to Reviewer 2: Thank you for your careful analysis on this topic. In the specific article under discussion, the focus is on nursing care for the patient. It is important to highlight that a greater understanding of hope in the context of nursing care by nurses is imperative. This understanding allows for greater ease in the selection of diagnoses, interventions, and assessment instruments in hope. This review did not intend to explore the meaning of experiences of hope from the personal perspective of nurses. For this reason, the title proposed by the authors contemplates the concept of «nursing care».

Reviewer 2: Material and Methods Section_"Figure 1 PRISMA flow chart: please reconfirm the number. In the identification step, there are 320 plus 2 articles and then records after duplicates are removed (n=234). Why did the screening step show N=235?”

Response to Reviewer 2: The error has been rectified in figure 1 PRISMA flow chart.

Reviewer 2: Material and Methods Section_" The format of Figure 2 needs to adjust (Line 187-188, 197-220, 203-205).”

Response to Reviewer 2: Figure 2 has been adjusted. Thanks for the suggestion.

Reviewer 2: Material and Methods Section_"Please explain how to form the results of “hope is a dynamic emotional state” and “hope is central to the process of adaptation” in Figure 2”.

Response to Reviewer 2: The explanations how to form the results of "hope is a dynamic emotional state" and "hope is central to the process of adaptation" are elucidated in the concluding section of the results.

(…) “hope is a dynamic emotional state” and “hope is central to the process of adaptation”. These dimensions highlight that hope is not a static emotion (dynamic emotional state); it changes over time and plays a central role in how individuals adapt to and navigate the challenges and uncertainties of life (adaptative process). For this reason, it serves as a motivating and adaptive force that can help people persevere and thrive in various situations.

Reviewer 2: Results_"Suggest: Authors need to explain in detail how to form the results of antecedents, attributes and consequences of hope in this paper. There is no information provided.”

Response to Reviewer 2:  A detailed explanation how to form the results has been added to the text.

The authors compared with classifications (NANDA-I diagnoses and ICNP® axis and similar terms) and propose new definitions for the hope diagnose and a new term for ICNP® (See in the results section Figure 1 - Proposed update of the definition of hope in the ICNP®).

Reviewer 2: Results_ “Suggest: Table S1 should be in the context because it provides crucial information of this manuscript.”

Response to Reviewer 2:  The authors agree and thank you for the suggestion. The following has been included TableS1 as Table 1 in the results section.

Reviewer 2: Results_ “Generation of new knowledge related to hope from this manuscript is critical. add and explain in detail what the updated definition of hope is compared with the ICNP. The information provided here is not sufficient.”

Response to Reviewer 2:  A comprehensive explanation of the revised definition of hope has been incorporated into the text. Additionally, an illustrative figure (Figure 3 - Proposed update of the definition of hope in the ICNP®) has been included to facilitate the reader's clear differentiation between the current ICNP definition and the proposed definition. We greatly appreciate your analysis and suggestions.

Reviewer 2: Results_ “Line 210: mention these qualities suggest…. Please clarify the meaning of these qualities here.”

Response to Reviewer 2: Sentence clarified.

Data indicates that hope is a comprehensive concept that requires support to achieve positive outcomes.

Reviewer 2: Results_ “Line 217-218: if the direct citation “…” is needed here, authors should explain their meaning.”

Response to Reviewer 2: Our meaning explained. 

It is crucial to consider and prioritize these aspects, as they play a vital role in the overall well-being and satisfaction of patients or individuals receiving healthcare services.

Reviewer 2: Results_ “Line242-243: Please explain the results of faith and spirituality, and winning position. The previous consequences of hope have a verb such as developing a new feeling of safety and formatting new and realistic goals.  Only using the terms of faith and spirituality, and winning position as the consequences of hope do not make sense.”

Response to Reviewer 2:   Faith and the perception of the transcendent emerge as a consequence, as they become a resource for individuals, particularly in times of uncertainty and vulnerability. People turn to and implement this resource with greater intentionality and purpose. We incorporated this explanation into the article.

Reviewer 2: Discussion_ “Suggestion: Given a heading of antecedents, attributes and consequences of hope and discuss in each would be great.”

Response to Reviewer 2: The antecedents, attributes and consequents of the concept have been listed in the text, in continuity with the numbering in the text (3.1; 3.2; 3.3).

Reviewer 3 Report

I would like to congratulate the authors for the manuscript they present, but in my opinion it can be improved. My comments are organized below for your consideration. I hope my comments are useful for the study author(s) and editorial staff.

#1 Introduction:

This section seems adequate for the objectives of the study, but the theoretical background could be improved by citing more articles less than 5 years old.

The citations (6; 20; 21; 22) are all from articles published by the authors, although I consider your knowledge in this area to be excellent, in order to enrich the literature of your manuscript and support diversity, I advise you to cite more articles from other renowned scholars in this field.

#4 Discussion

The discussion of the results, in general, is not supported by recent studies, as the vast majority of the studies referred to are over 5 years old, so I think it would be important to cite more recent studies, less than 5 years old.

# References

- The authors should update the references, as around 49% are more than 5 years old.

Author Response

We wish to express our sincere appreciation for the invaluable suggestions offered by you in your role as a reviewer for our article titled “What do we know about hope in nursing care?: A systematic synthesis review of concept analysis studies.” We are committed to providing a point-by-point response to your feedback. On behalf of the authors, we extend our heartfelt thanks for granting us the opportunity to review and provide clarification on these crucial matters.

REVIEWER 3

Reviewer 3: Introduction Section:This section seems adequate for the objectives of the study, but the theoretical background could be improved by citing more articles less than 5 years old.”

Response to Reviewer 3: Given the nature of the study, in an introductory context, the authors followed the description of the evolution of the concept of hope in the nursing discipline, which justifies the need to use studies/authors with publications dating back more than 5 years.

Reviewer 3: Introduction Section:The citations (6; 20; 21; 22) are all from articles published by the authors, although I consider your knowledge in this area to be excellent, in order to enrich the literature of your manuscript and support diversity, I advise you to cite more articles from other renowned scholars in this field.”

Response to Reviewer 3: Thank you for this analysis and reinforcement. We agree with the suggestion and have therefore diversified the bibliographic sources throughout the manuscript.

Reviewer 3: Discussion:The discussion of the results, in general, is not supported by recent studies, as the vast majority of the studies referred to are over 5 years old, so I think it would be important to cite more recent studies, less than 5 years old.”

Response to Reviewer 3: Thank you for your suggestion.  However, as this is a "Discussion" chapter, the authors used the theoretical sample included in the review and which met the study's inclusion criteria. The years of publication reflect each of the integrated concept analyses, which justifies the authors' choice not to add other bibliographic sources in this section of the manuscript.

Reviewer 3: References:The authors should update the references, as around 49% are more than 5 years old.”

Response to Reviewer 3: Thank you for your suggestion. We have reinforced the entire manuscript with more recent scientific evidence (over 5 years old), although these are primary studies not characterized as concept analysis.

Reviewer 4 Report

Dear authors,

The question asked by the authors in the title (and the review question) seems pertinent, as I have made a preliminary search through PubMed and there do not seem to be many publications that address using the terms "Hope" and "Nursing Care". On the other hand, the authors do not report registration of the research protocol in PROSPERO or any other repository.

Thus, in the methods section the sentence "A narrative synthesis was used as a methodological strategy to clarify and construct a conceptual model of hope [27]." introduces doubts on the methodology of the type of review developed. (Narrative synthesis corresponds to subheading Data extraction).

These three aspects, together with the fact that the inclusion criteria do not place any restrictions on the type of studies, make me think that the type of review could be a scoping review. Although scoping review protocols cannot be published in PROSPERO, these protocols can be published in repositories such as the Open Science Framework (OSF).

Keywords should correspond to MesH terms for correct indexing: The keyword "Holistic Care" is not a MesH term ("Holistic Nursing" or "Holistic Healh" are alternative MesH if you consider them appropiate). The correct MesH for "Nursing Diagnoses" is "Nursing Diagnosis". "Review Literature as Topic" is the correct MesH for keyword "Review Literature" (alternative keyword is "Systematic review"). "Therapeutic Relationships" is not a MesH term.

Abstract: Is informative. They should correct the databases they have searched (Pubmed is the search engine used for the Medline database).

You must corrrect the sentence: "The search terms “hope” AND “concept” AND “analysis” AND “Nurs*” in the title were used to identify potential papers". Is no correct the term: "Nur*". This corresponds to a search strategy with "truncation", which should be explained in the methods section.

Introduction: Correct.

Methods: You must cite the PRISMA 2020 statement (Page et al, 2020) (https://www.equator-network.org/reporting-guidelines/prisma/).

PRISMA statement indicates the framework followed for reporting the research report. In addition to these criteria, the framework in which the review was conducted must be specified (e.g. JBI).

Inclusion criteria in relation to study design should be indicated (in addition to patients and settings). Specify if filters have been established for the year of publication.

Information sources: According PRISMA Checklist you must provide the full search strategies for all databases, including any filters and limits used (you can provide it as supplementary file for each database).

Figure 1. In my opinion, it should be presented as the first result in the results section, specifying the process of elimination of duplicates, screening of records by title and abstract, screening of full text records and studies included in the review (In the results statement, the first paragraph describes this process). The flow diagram is not correctly described, it should specify each of the steps (e.g. how many records have been removed and how many remain after removal?). You can see the correct diagram if you includes searches of databases, registers and other sources, such us citation searching at this place: http://prisma-statement.org/prismastatement/flowdiagram.aspx.

Selection process: Specify the methods used, critical appraisal tool, to decide whether a study met the inclusion criteria.

Results: Correct (except flow diagram).

Discussion: Appropiate.

Conclusión: Correct.

Author Response

We wish to express our sincere appreciation for the invaluable suggestions offered by you in your role as a reviewer for our article titled “What do we know about hope in nursing care?: A systematic synthesis review of concept analysis studies.” We are committed to providing a point-by-point response to your feedback. On behalf of the authors, we extend our heartfelt thanks for granting us the opportunity to review and provide clarification on these crucial matters.

REVIEWER 4

Reviewer 4: “The question asked by the authors in the title (and the review question) seems pertinent, as I have made a preliminary search through PubMed and there do not seem to be many publications that address using the terms "Hope" and "Nursing Care". On the other hand, the authors do not report registration of the research protocol in PROSPERO or any other repository.”

Response to Reviewer 4: Thank you for your attentive and rigorous eye. The registration was not carried out on Prospero or any other resource, as this is not a systematic review, but a synthesis of studies that have carried out conceptual analyses on the concept of hope. We have made this clear in the methodological chapter of the manuscript.

Reviewer 4: “Thus, in the methods section the sentence "A narrative synthesis was used as a methodological strategy to clarify and construct a conceptual model of hope [27]." introduces doubts on the methodology of the type of review developed. (Narrative synthesis corresponds to subheading Data extraction).”

Response to Reviewer 4: Thank you for your suggestion. This has been corrected in the manuscript.

Response to Reviewer 4:These three aspects, together with the fact that the inclusion criteria do not place any restrictions on the type of studies, make me think that the type of review could be a scoping review. Although scoping review protocols cannot be published in PROSPERO, these protocols can be published in repositories such as the Open Science Framework (OSF).”

Response to Reviewer 4: As this is a literature review that does not use review methodologies such as scoping review or systematic review, it was not registered on the OSF platform, as registration is not mandatory for this type of review.

Reviewer 4: “Keywords should correspond to MesH terms for correct indexing: The keyword "Holistic Care" is not a MesH term ("Holistic Nursing" or "Holistic Health" are alternative MesH if you consider them appropiate). The correct MesH for "Nursing Diagnoses" is "Nursing Diagnosis". "Review Literature as Topic" is the correct MesH for keyword "Review Literature" (alternative keyword is "Systematic review"). "Therapeutic Relationships" is not a MesH term.”

Response to Reviewer 4: We appreciate the valuable correction. We have incorporated only MeSH terms into the list of keywords.

Keywords: Holistic Health; Hope; Nursing Diagnosis; Nursing Theory; Review Literature; Nurse-Patient Relations; Therapeutical Approaches. 

Reviewer 4: Abstract: “Is informative. They should correct the databases they have searched (Pubmed is the search engine used for the Medline database).”

Response to Reviewer 4: We removed Pubmed database from the abstract.

Reviewer 4: “You must corrrect the sentence: "The search terms “hope” AND “concept” AND “analysis” AND “Nurs*” in the title were used to identify potential papers". Is no correct the term: "Nurs*". This corresponds to a search strategy with "truncation", which should be explained in the methods section).”

Response to Reviewer 4: Thank you very much for this request for clarification. We have corrected the description as follows:

The search terms used are “hope” AND “concept” AND “analysis” AND “nurs*”. The “truncation” aim recovers singular and plural variations or differences in spelling and word endings. Regarding the term nursing, this aspect is quite significant and had to be considered in the search strategy.

Reviewer 4: Methods: “You must cite the PRISMA 2020 statement (Page et al, 2020) (https://www.equator-network.org/reporting-guidelines/prisma/).”

Response to Reviewer 4: The authors use the reference [28], Shamseer, L., et al. Preferred reporting items for systematic review and meta-analysis protocols (PRISMA-P) 2015: elaboration and explanation. BMJ (Clinical research ed.), 2015, 350, g7647. doi: https://doi.org/10.1136/bmj.g7647. The following additional information has been added to the figure's 1:

Figure 1. PRISMA flow chart of the study selection [28] (Adapted to https://www.equator-network.org/reporting-guidelines/prisma/).

Reviewer 4: “PRISMA statement indicates the framework followed for reporting the research report. In addition to these criteria, the framework in which the review was conducted must be specified (e.g. JBI).”

Response to Reviewer 4: The framework of the review follows the PRISMA orientation.

Reviewer 4: “Inclusion criteria in relation to study design should be indicated (in addition to patients and settings). Specify if filters have been established for the year of publication).”

Response to Reviewer 4: Suggestions added.

This review uses theoretical-methodological models describing the concept of hope in nursing practice. Peer-reviewed articles with fully accessible Portuguese or English text were considered. As we aimed to include a historical perspective of the concept, no re-striction upon time of publication was set. We included concept analyses covering all ages, settings, contexts and study designs. No filters have been established for the year of publication.

Reviewer 4: Information sources: “According PRISMA Checklist you must provide the full search strategies for all databases, including any filters and limits used (you can provide it as supplementary file for each database).”

Response to Reviewer 4: This information has been considered in Search and selection strategy section.

Reviewer 4: Figure 1. “In my opinion, it should be presented as the first result in the results section, specifying the process of elimination of duplicates, screening of records by title and abstract, screening of full text records and studies included in the review (In the results statement, the first paragraph describes this process). The flow diagram is not correctly described, it should specify each of the steps (e.g. how many records have been removed and how many remain after removal?). You can see the correct diagram if you includes searches of databases, registers and other sources, such us citation searching at this place: http://prisma-statement.org/prismastatement/flowdiagram.aspx.).”

Response to Reviewer 4: Thank you for your suggestions, which we have included in figure 1.

Reviewer 4: Selection process: “Specify the methods used, critical appraisal tool, to decide whether a study met the inclusion criteria.”

Response to Reviewer 4:  As this is not a systematic review, the authors argue that it is not mandatory to analyse the methodological quality of the articles included.

Round 2

Reviewer 2 Report

The authors have revised accordingly. Please clarify and revise the following items  before publication.

1.          Line 40 and 41:  There is only one sentence in one paragraph. Suggest linking with the content from Line 42.

2.          Please adjust the format from Line 224-241.

3.          Please confirm: The format of Figure 3 – Proposed update of the definition of hope in the ICNP®. What do you mean by Hope (10009095) in Figure 3?

Author Response

Dear Reviewer,

Once again, we greatly appreciate all the reviewers' suggestions for the manuscript. The authors respond in continuity point by point to all questions. We are convinced that the manuscript will benefit in quality and clarity. Thank you very much.

REVIEWER 2

Reviewer 2: " Line 40 and 41:  There is only one sentence in one paragraph. Suggest linking with the content from Line 42."

Response to Reviewer 2: Thank you for the suggestion. We have now linked lines 40 and 41 with line 42.

Reviewer 2: " Please adjust the format from Line 224-241."

Response to Reviewer 2:  The adjustment has been made in Figure 2 (lines 224-241).

Reviewer 2: " Please confirm: The format of Figure 3 – Proposed update of the definition of Hope in the ICNP®. What do you mean by Hope (10009095) in Figure 3?"

Response to Reviewer 2: In the context of the International Classification for Nursing Practice (ICNP), "Hope (10009095)" is a code or identifier used to represent the concept related to Hope within the framework of nursing practice. ICNP is an international standard for classifying nursing diagnoses, interventions, and outcomes to facilitate standardized communication and documentation among nurses. In any case, we've altered Figure 3 to Table 3 to make it more legible. The changes made have been marked in the manuscript.  

Reviewer 4 Report

Dear authors, thank you clarifying that the type of review does not correspond to a systematic or scoping review; this is a methodological limitation that should be reflected in the discussion, so that the results and conclusions will be considered in terms of the research design. The title should not indicate that it is a systematic review. The aim of the research should also not describe systematically.

Supplemenary table is the same: "Table S1 Synthesis of concept analysis studies on hope" in the manuscript and in the supplementary file. To improve the rigour of the process, search strategies should be included as a supplementary file.

Methods: Search strategy: Delete PubMed on page 4 (line 158).

Check the sentence: "[The key Medical Subject Headings (MeSH) terms used were ‘hope*’, ‘concept’, ‘analyses’, ‘nurs*’, combined with the Boolean operator AND". You cannot affirm that you have used MeSH terms as none of them are MeSH terms; "analyses" and "concept" are keywords using free terms, but no descriptor terms, and "nur*" and "hope*" are truncations corresponding to the search strategy.

In my opinión, the flow chart should indicate in the first step the number of duplicate records that have been deleted (¿n= 87?) and in the second step the number of records remaining after deleting the duplicates (n= 235).

I consider the diagram to be the first result, so it should be moved to this section, after the first results paragraph.

On page 20 (lines 285-286) you mention NANDA-I, but this corresponds to the acronym for NANDA International (since this is the only time it appears in the text, it is not necessary to use the acronym). 

Conclusions: Delete the word "systematic" where it occurs.

Author Response

Dear Reviewer,

Once again, we greatly appreciate all the suggestions for the manuscript. The authors respond in continuity point by point to all questions. We are convinced that the manuscript will benefit in quality and clarity. Thank you very much.

REVIEWER 4

Reviewer 4: "Dear authors, thank you for clarifying that the type of review does not correspond to a systematic or scoping review; this is a methodological limitation that should be reflected in the discussion so that the results and conclusions will be considered in terms of the research design. The title should not indicate that it is a systematic review. The aim of the research should also not describe systematically."

Response to Reviewer 4:  Thank you for the correction. It was an oversight by the authors not to make these changes in the previous revision process. The title of the manuscript has been changed from "What Do We Know about Hope in Nursing Care?: A Systematic Synthesize Review of Concept Analysis Studies" to: "What Do We Know about Hope in Nursing Care?: A Synthesize Review of Concept Analysis Studies". The objective has been corrected from "This study aims to systematically synthesize research regarding concept analyses of hope in nursing care and to propose an evidence-based update of the definition of hope in the International Classification for Nursing Practice (ICNP®) taxonomy" to "This study aims to synthesize research regarding concept analyses of hope in nursing care and to propose an update of the definition of hope in the International Classification for Nursing Practice (ICNP®)."

Reviewer 4: "Supplementary table is the same: "Table S1 – Synthesis of concept analysis studies on hope" in the manuscript and in the supplementary file. To improve the rigour of the process, search strategies should be included as a supplementary file."

Response to Reviewer 4: The search strategies have been included as a supplementary file, not Table S1. Thank you for the correction.

Reviewer 4: "Methods: Search strategy: Delete PubMed on page 4 (line 158)."

Response to Reviewer 4: PubMed has been removed on page 4 (line 158).

Reviewer 4: "Check the sentence: "[The key Medical Subject Headings (MeSH) terms used were 'hope*', 'concept', 'analyses', 'nurs*', combined with the Boolean operator AND". You cannot affirm that you have used MeSH terms as none of them are MeSH terms; "analyses" and "concept" are keywords using free terms, but no descriptor terms, and "nur*" and "hope*" are truncations corresponding to the search strategy."

Response to Reviewer 4: We appreciate the clarification. The sentence should accurately represent the use of MeSH terms and free-text keywords. Here's a revised version: "The key Medical Subject Headings (MeSH) terms used included 'hope' combined with the free-text keywords 'concept' and 'analyses,' as well as the truncations 'nur*' and 'hope*' with the Boolean operator AND."

Reviewer 4: "In my opinión, the flow chart should indicate in the first step the number of duplicate records that have been deleted (¿n= 87?) and in the second step the number of records remaining after deleting the duplicates (n= 235)."

Response to Reviewer 4: Thank you for your suggestion. We have incorporated the number of duplicates that were removed as a step in the flow chart.

Reviewer 4: "I consider the diagram to be the first result, so it should be moved to this section, after the first results paragraph."

Response to Reviewer 4: Thank you for your suggestion. The suggested change has been made. The proposed changes were made and marked respectively in the manuscript.

Reviewer 4: "On page 20 (lines 285-286) you mention NANDA-I, but this corresponds to the acronym for NANDA International (since this is the only time it appears in the text, it is not necessary to use the acronym)."

Response to Reviewer 4: Thank you for your suggestion. We have revised the acronym NANDA-I to NANDA International.

Reviewer 4: "Conclusions: Delete the word "systematic" where it occurs."

Response to Reviewer 4: We have revised the manuscript and removed the word "systematic" from all the text.
